# Evaluation of the Sealing Ability and Bond Strength of Two Endodontic Root Canal Sealers: An In Vitro Study

**DOI:** 10.3390/dj10110201

**Published:** 2022-10-26

**Authors:** Manuel Marques Ferreira, José Pedro Martinho, Inês Duarte, Diogo Mendonça, Ana Catarina Craveiro, Maria Filomena Botelho, Eunice Carrilho, Carlos Miguel Marto, Ana Coelho, Anabela Paula, Siri Paulo, Nuno Chichorro, Ana Margarida Abrantes

**Affiliations:** 1Institute of Endodontics, Faculty of Medicine, University of Coimbra, 3000–354 Coimbra, Portugal; 2Center for Innovative Biomedicine and Biotechnology (CIBB), University of Coimbra, 3000–354 Coimbra, Portugal; 3Clinical Academic Center of Coimbra (CACC), 3000–354 Coimbra, Portugal; 4Coimbra Institute for Clinical and Biomedical Research (iCBR) Area of Environment Genetics and Onco-Biology (CMAGO), Faculty of Medicine, University of Coimbra, 3000–354 Coimbra, Portugal; 5Institute of Biophysics, Faculty of Medicine, University of Coimbra, 3000–354 Coimbra, Portugal; 6Institute of Integrated Clinical Practice, Faculty of Medicine, University of Coimbra, 3000–354 Coimbra, Portugal; 7Institute of Experimental Pathology, Faculty of Medicine, University of Coimbra, 3000–354 Coimbra, Portugal; 8Institute of Nuclear Sciences Applied to Health, Faculty of Medicine, University of Coimbra, 3000–354 Coimbra, Portugal

**Keywords:** AH-Plus^®^, bond strength, GuttaFlow Bioseal^®^, push-out test, root canal sealers, sealing capacity

## Abstract

Background: Obturation represents a critical step in endodontic treatment, which relies on a core material and a sealer. This study aims to evaluate the sealing ability and bond strength to the root canal walls of an epoxy resin-based sealer (AH-Plus^®^, Dentsply Sirona, Johnson City, TN, USA) and a bioceramic sealer (GuttaFlow Bioseal^®^, Coltène/Whaledent, GmbH + Co. KG, Langenau, Germany). Methods: Thirty-eight maxillary anterior teeth with single roots and identical round sections were separated into two experimental groups according to the root canal sealers used, namely, G1 = AH-Plus^®^ and G2 = GuttaFlow Bioseal^®^, and two control groups, specifically, G3 = the negative control and G4 = the positive control. The sealing capacity was measured by the penetration of the radioactive isotope ^99^mTc. For the push-out test, the compressive force test was performed in a universal machine and the force was applied by exerting pressure on the surface of the material to be tested in the apical to the coronal direction and using three test points with different diameters. Results: GuttaFlow Bioseal^®^ exhibited superior sealing ability compared to AH-Plus^®^ (*p* = 0.003). Regarding the bond strength, AH-Plus^®^ provided higher adhesion values than GuttaFlow Bioseal^®^ in the three sections of the tooth root (*p* = 0.001). Conclusions: GuttaFlow Bioseal^®^ had significantly better sealing ability than AH-Plus^®^ but lower adhesion values in the three zones of the root canal, with statistically significant differences between the groups. However, it is important to note that for the action of endodontic sealers to be maximized, the root-filling technique must be most appropriate.

## 1. Introduction

The success of an endodontic treatment depends on the cleaning, modelling, and 3D obturation of the root canal system [1,2]. Root canal filling, a crucial phase, aims to prevent the progression of bacteria and fluid entry from the oral cavity into the periapical tissues, trap bacteria that have resisted the intracanal instrumentation/irrigation phase, and obstruct the entry of periradicular exudates [2,3,4,5].

Regarding the canal-sealing procedure, several materials have been advocated, such as core materials and root canal sealers. The obturation materials must have specific physical, biological, and handling properties [6,7]. They must establish a bond between the core material and the root canal’s dentine walls, and fill the areas of the canal’s anatomy inaccessible to solid materials to prevent the leakage of fluids and bacteria [5,8,9].

Gutta-percha is the most frequently used core material, but this material does not adhere to the dentinal walls of the canal [1,8,10]. So, diverse materials are available for this purpose, such as zinc oxide and eugenol-based, calcium hydroxide-based, glass ionomer-based, resin epoxy-based, silicone-based, and more recently bioceramic-based sealers [6,11,12,13].

The epoxy resin-based root canal sealers, combined with gutta-percha, have been the most used materials in endodontic treatments [2]. However, other techniques and materials with different physicochemical and biological properties have been developed [2]. The epoxy resin-based sealers possess excellent physical properties, such as a slow setting reaction, low solubility, high flow rate, low volumetric polymerization contraction, and adaptation to the canals’ dentine walls [10].

As previously related, bioceramic materials have emerged as a new option in dentistry [14]. The use of bioceramic materials as endodontic sealers presents several advantages, such as biocompatibility, osteoconductivity, good sealing capacity, adhesion, and good radiopacity, in addition to containing calcium phosphate in their composition [6,14]. Biocompatibility prevents rejection by the surrounding tissues and the calcium phosphate improves their setting properties and results in a chemical composition and crystal structure similar to tooth and bone hydroxyapatite [6,15]. All these characteristics led to the widespread use of these materials in endodontics [14]. The greatest disadvantage of their use is the difficulty in removing the material from the root canal if a re-treatment is necessary or during the preparation of the canal for a dental post [6]. GuttaFlow Bioseal^®^ (Coltene/Whaledent, GmbH + Co. KG, Germany), which has gutta-percha and calcium silicate particles in its composition, was recently introduced into the market [16]. This sealer is said to release natural repair constituents and aid the regeneration of periapical tissues upon contact with biological tissues [2].

This in vitro study aims to evaluate the sealing ability and the bond strength of an epoxy resin-based sealer (AH-Plus^®^, Dentsply Maillefer, Ballaiques, Switzerland) and a bioceramic sealer (GuttaFlow Bioseal^®^). The null hypothesis was that no difference exists between the two materials regarding their sealing ability and bond strength.

## 2. Materials and Methods

### 2.1. Sample Collection and Preparation

This study was conducted according to the principles of the Declaration of Helsinki (10th version, October 2013) (www.wma.net, accessed on 1 September 2018) and in accordance with the Medical Research Involving Human Subjects Act (WMO). The protocol was approved by the Ethics Committee of the Faculty of Medicine of the University of Coimbra, Portugal (ref^a^ CE-107/2017).

The sample was gathered from patients that showed an indication requiring the extraction of their affected teeth, performed in the Dental Medicine Department, Faculty of Medicine, University of Coimbra, Portugal.

Maxillary monoradicular anterior teeth were selected, with a single canal (Weine type I) and identical round sections, no root caries, closed apex, no previous endodontic treatment, no root fractures, and no resorption (observed using the magnification). After the teeth were collected, the periodontal ligament was removed with a Gracey curette, and the teeth were disinfected with 3% NaOCl and stored in chloramine T at 4 °C until used. The canal anatomy was confirmed by performing periapical radiographs with different angulations.

Later, the crown was separated from the root via sectioning with high-velocity carborundum discs perpendicular to the long axis of the root to obtain 14 mm long root segments. The root canal preparation was performed using the ProTaper^®^ Universal (Dentsply Maillefer, Ballaiques, Switzerland) rotary files in the sequence S2, F1, F2, and F3 (300× *g* rpm). The working length was established by inserting an ISO size K-file 10 (Dentsply Maillefer, Ballaigues, Switzerland) until its exit at the apex was visible and 1 mm was reduced from the length of the file. After the use of each instrument, the canal was irrigated with 2 mL of 3% NaOCl (CanalPro^®^, Coltène/Whaledent Inc., Cuyahoga Falls, OH, USA) and the apex permeabilized with a k10 file. The final irrigation was performed with 1 mL of 17% EDTA (Coltène/Whaledent Inc., Langenau, Germany, D-89122) for 1 min to remove the smear layer, followed by 70% ethanol and neutralization with 2 mL of saline solution. Before obturation, the canals were dried with paper points (Dentsply Maillefer, Ballaigues, Switzerland).

After root canal preparation, the samples were divided into four groups: G1 (AH-Plus^®^, *n* = 15), G2 (GuttaFlow Bioseal^®^, *n* = 15), G3 (Negative Control, *n* = 4), and G4 (Positive Control, *n* = 4). In groups G1, G2, and G3, the samples were filled with the single-cone technique, using a Protaper Universal F3 (Dentsply Maillefer, Ballaiques, Switzerland) gutta-percha cone. To cement the core material to the canal walls, AH-Plus^®^ sealer was used in the G1 and G3 groups, and GuttaFlow Bioseal^®^ sealer in the G2 group. Group 3 was used as the negative control, as detailed in the following section. Group 4 was used as positive control; the canals were not obturated with any root canal filling materials and were left empty.

The cones were inserted slowly to the established working length, sectioned with a heated instrument at cervical level, and slightly condensed with a plugger. Excess gutta-percha was removed with a heated instrument. In the G4 group, the samples were not filled. The same operator performed all root canal preparations and obturations.

For setting the sealers, the roots were kept at room temperature for 48 H in an atmosphere of 100% of humidity.

The endodontic sealers used in root canal filling and their composition are described in Table 1.

### 2.2. Sealing Ability Evaluation

The root surfaces of each tooth from G1, G2, and G4 groups were covered with varnish, except the apical 3 mm root. For the G3 group, the negative control, the roots were completely covered with varnish to completely seal the root, and in G4, the positive control, the empty root was also varnished along the external surface of the root, except in the 3 mm apical root.

All roots were immersed in a 99 m technetium solution (^99m^Tc) for 3 h. After this time, the varnish was removed and the counts for each tooth were obtained in a gamma-camera (GE 400 AC, Milwaukee, WI, USA). For each tooth, a static image was acquired for 3 min, and regions of interest (ROIs) were drawn on each tooth, with a 512 × 512 matrix size, and total, mean, and minimum counts *per* minute (cpm) were obtained. The entire procedure was carried out by a nuclear medicine specialist.

### 2.3. Bond Strength Evaluation

To evaluate the bond strength, the push-out test was chosen and G1 (*n* = 15) and G2 (*n* = 15) were considered. Thus, the 30 samples were cased in Tab 2000 (Kerr, Scafati, Italy) and then 0.5–2.3 mm thick portions were sectioned with a high-precision machine (Exact 310 CP, Norderstedt, Germany), perpendicular to the long axis of the root, in three zones of the root: apical, middle, and cervical. The sections were made between a distance of 2 mm from the apex and 2 mm under the cervical zone. For each of these zones, 12 samples were selected, yielding a total of 72 samples, 36 for each group. The segments were analysed with a microscope (Nikon, SMZ-1500, Tokyo, Japan) at 30× and 40× magnifications and photographed with a coupled digital camera (Nikon, High resolution, 12.6-megapixel DXM-1200C, Tokyo, Japan) using ACT-1C software (Nikon, DXM-1200C, Tokyo, Japan). Then, the radii of the samples were determined using the software ImageJ 1.30 (Image Processing and Analysis in Java; National Institutes of Health, Bethesda, MD, USA) (Figure 1A,B).

The push-out test was performed with a universal machine (Shimadzu AG-I, Shimadzu Corporation, Kyoto, Japan).

The compressive force was applied by exerting pressure on the material surface from the apical to the coronal direction to avoid movement limitation and using three test points with different diameters: 0.9 mm, 0.7 mm, and 0.5 mm. The loading speed was 1 mm/min until the failure of the adhesive interface. The maximum force applied to the material at the time of its displacement was recorded in Newtons (N), for which the application of force ceased when the measurement reached zero. Then, the tension required to move the material was calculated in mega Pascals (MPa), using the following formula [17,18,19,20,21].

Push-out bond strength (MPa) = Maximum load (Newton)/Cement adhesive area to root canal (mm^2^): $Radial\;canal\;adhesive\;area\;\left(mm^{2}\right)=\pi \left(R+r\right)h$, where π = 3.14, R denotes the radicular canal radius in the most coronal region, r is the radius of the canal in the most apical region, and h is the thickness of the sample in millimeters.

### 2.4. Statistical Analysis

For the sealing ability evaluation, the description of the evaluated groups was performed using the mean, standard deviation (SD), and maximum and minimum values obtained. To assess the differences between the groups, the Kruskal–Wallis test was used after the assumption of normality was verified through the Shapiro–Wilk test. Post hoc tests (Dunn–Sidak) were performed with correction for multiple comparisons to assess the differences between each pair of groups.

For the bond strength analyses, first, the outliers were eliminated by calculating mean ± 3SD. Afterwards, the descriptive statistics were defined, which comprised the median central tendency measures, 25th and 75th percentiles, mean (standard error and 95% confidence interval), amplitude (minimum and maximum value), and standard deviation as a measure of dispersion. To statistically assess the differences between the two materials, the Mann–Whitney test was used. This test was selected after the normality assumption of the variables was tested via the Shapiro–Wilk test. The effect size was determined as a Cohen’s r value as proposed by Fritz et al. [22]. In addition, comparisons were made between the materials per tooth root zone (cervical, middle, and apical) using the Mann–Whitney test.

Statistical significance was set at *p* < 0.05. All analyses were performed using the statistical software Statistical Package for the Social Sciences, version 23 (SPSS Inc., IBM Company, Armonk, NY, USA).

## 3. Results

### 3.1. Sealing Ability Evaluation

The descriptive statistics (mean, standard deviation, and maximum and minimum values) are shown in Table 2.

Regarding the comparison between the groups, there were statistically significant differences between G3 and G4 (*p* = 0.010), G2 and G4 (*p* = 0.001), and G1 and G2 (*p* = 0.003). Between G1 and G4, no statistically significant differences were observed (*p* = 0.695).

### 3.2. Bond Strength Evaluation

Table 3 summarizes the descriptive statistics (amplitude, mean, median, and standard deviation) of the two experimental groups, AH-Plus^®^ and GuttaFlow Bioseal^®^. For this analysis, three areas (cervical, middle, and coronary) were considered. There was homogeneity regarding the most cervical radius, the most apical radius, and the thickness of the samples. However, the force applied to displace the endodontic cement was higher in the AH-Plus^®^ group (20.244 MPa) compared to the GuttaFlow Bioseal^®^ one (4.880 MPa).

Table 4 presents the main effects between the two sealers considering all the variables of the study without considering the different areas of the tooth roots. Statistically significant differences were observed regarding force (F) (*p* = 0.001), the thickness of the sample (*p* < 0.001), and tension (*p* < 0.001).

Table 5 presents the means and the standard deviations per endodontic cement. In this table, the results from each sample area (cervical, middle, and apical) and the effects the cement had on them are presented.

Statistically significant differences were found regarding the thickness of the cervical and middle zones (*p* = 0.001), and the strength of the apical zone (*p* = 0.001). There were statistically significant differences in the apical zone (*p* = 0.001) and the middle zone (*p* = 0.003) regarding the tension.

## 4. Discussion

An ideal endodontic sealer should provide excellent root canal sealing, dimensional stability, sufficient setting time to ensure working time, insolubility against tissue fluids, adequate adhesion to canal walls, and biocompatibility [7,23,24,25,26].

In this study, we observed that GuttaFlow Bioseal^®^ provided a statistically significantly higher sealing ability than AH-Plus^®^ but lower adhesion values in the three zones of the root canal.

To assess the sealing ability of endodontic sealers, radioisotope infiltration was used. This is a method that has several advantages, wherein the most notable is the fact that it is a non-destructive method that is highly sensitive for the evaluation of micro-infiltration [27]. In addition, it is not absorbed by the dentin matrix or by the hydroxyapatite crystals [28] and does not prevent future use of the samples. Another advantage of radioisotope infiltration is that it enables the evaluation of the molecules’ infiltration. Passive ^99^mTc infiltration occurs through capillarity, so the fact that a filling enables the passage of this molecule does not mean that it allows bacteria to re-contaminate the root canals, since the bacteria are larger than the molecules [29]. Other methods exist to evaluate apical microleakage, such as the use of dyes. Although this method is the most commonly used, it is insufficiently sensitive and some dyes react with dentin, which can lead to unfavorable results [30].

The results regarding the sealing ability can be attributed to the different compositions and properties of the two endodontic sealers under study, namely, the hydrophobicity and shrinkage of AH-Plus^®^ [31]. Since epoxy resin-based sealers tend to retract during setting, there may be a disintegration of the sealer in the root canal walls, increasing gaps between this interface [32]. In addition, AH-Plus^®^ is a hydrophobic sealer, which can lead to a poor adaptation of this sealer to the canal wall [33]. On the other hand, the results regarding GuttaFlow Bioseal^®^, a hydrophilic sealer, can be attributed to its better flowability and ability to expand slightly upon setting [34,35]. In addition, GuttaFlow Bioseal^®^ forms a physical bond between the calcium silicate and the dentin surface [16]. The presence of apatite interface deposits, calcium release, low solubility, and the alkalizing activity of the calcium ions and phosphate ions stimulate the development of a superficial layer of calcium phosphate, which can fill in the voids and improve the sealing ability [16,36].

The bond strength between the dentin and the obturation materials plays an important role in the success of endodontic procedures, as the increase in the adhesive properties to the dentin can lead to greater resistance to root fracture, a lower risk of microleakage, and the clinical longevity of a tooth subjected to an endodontic treatment [17,21,37]. Studies have shown that epoxy resin-based sealers have a greater bond strength to the root canal walls than most endodontic sealers [1,17,37,38]. These values are associated with the covalent bond between the epoxide ring (open circle) and the exposed sidechain amine groups in collagen, the low volumetric polymerization contraction, the low polymerization stress, and the long-term dimensional stability [1,10,17,38]. In our study, we obtained similar results, since AH-Plus^®^ provided higher bonding values than GuttaFlow Bioseal^®^ in the three zones of the tooth root [39,40]. These results may be justified by the fact that GuttaFlow Bioseal^®^ has silicone and gutta-percha particles in its composition, and studies indicate that gutta-percha and silicone do not adhere to the dentin surface [10,18,41]. However, GuttaFlow Bioseal^®^ has calcium silicate particles in its composition, which means that it does not contract, and this should enable a better adjustment of the sealer to the root canal system [21]. Although the degree of material contraction could be smaller, the inferior adhesion can explain the GuttaFlow Bioseal^®^ results.

Since adhesion to root canal walls is an essential property of root canal sealers, the push-out test is used to evaluate the bond strength between the obturation material and root dentin [1,9].The model is described as effective and reproducible, allowing the canal obturation materials to be evaluated, even when the adhesive forces are low [17,21,42]. Although we aimed to obtain the ideal conditions to perform the experiment, the push-out test presents some limitations with respect to the sample thickness, test tip diameter, and root canal diameter [42]. The sample must have a minimum thickness of 1–2 mm so that the formula used to calculate the adhesive forces will allow for the adhesive forces of the material to be determined [20,42]. Therefore, if the sample thickness is too small, there may be fractures in the material and/or dentin. A tip with a diameter (Dp) slightly lower than the canal diameter (Dc) is recommended so that the test force will be directed to the material–dentin interface [20]. However, the tip should not have a diameter much smaller than the diameter of the canal; otherwise, it will only drill the material and will not evaluate the adhesive force [20]. For these reasons, the Dp/Dc ratio should not exceed 0.85 to ensure that the results are closer to reality [20]. In this study, three test points were used to enable their adjustment to the diameter of the canal. Despite the important data obtained, the push-out test is a laboratory test and evaluates only one parameter of an endodontic sealer (adhesive force), so it should not be used as the only guideline for clinical decision making [43]. In addition, although the push-out test provides important information about the interaction between dentin and the root canal filling, it does not singularly represent the adhesion between them [43]. For example, with wet and dry moisture conditions, adhesive failure can occur. Additionally, some authors relate that this adhesive failure can also be related to the time of the test application [40].

In the present study, the single-cone obturation technique was used. Although this technique is simple and easily replicable, it is considered inferior to the others because it uses larger amounts of sealer. However, it is necessary to clarify that the gutta-percha cones were calibrated according to the diameter of the canal preparation; therefore, the volume of the sealer was minimized and distributed homogeneously throughout the root canal [34,35].

The use of both evaluation techniques lends strength to our results, as they allow us to understand the difference between the results of micro-infiltration tests and push-out bond strength tests.

Although the adhesive property and sealing ability values are only two variables in the selection of root canal sealers, the current study showed that, in terms of clinical situations, root canal sealers with higher results in the push-out test seem more advantageous when post-preparation is required [41].

Although in vitro studies are extremely important for the development of new techniques, more advantageous materials than the current ones, and technological advances, some limitations exist. First, we must be careful when extrapolating the results of in vitro studies to clinical situations, especially when several factors are involved. In addition, only teeth with straight canals were used, which is not observed in clinical practice. In teeth with curved canals, the root canal sealers will have more difficulty reaching the apical zone, leading to more voids within the root canal than those seen in teeth with straight canals. For this reason, the results of a similar study performed on teeth with curved canals may differ from those presented in this one. Although the study of endodontic sealers has shown favorable results, long-term assessments of their physicochemical properties, namely, a physical characterization of their porosity and long-term clinical observations, are necessary to evaluate the impact of different endodontic sealers on the outcome of root canal treatments.

## 5. Conclusions

Within the limitations of this study, GuttaFlow Bioseal^®^ had a significantly better sealing ability than AH-Plus^®^ but lower adhesion values in the three zones of the root canal. Therefore, the null hypothesis is rejected, as there are statistically significant differences between the groups.

However, it is important to note that in order to maximize the action of endodontic sealers, the root-filling technique must be appropriate to the materials used. This study demonstrates that a bioceramic sealer seems to have superior results when it is applied via the single-cone technique without warm vertical compaction.

## Figures and Tables

**Figure 1 dentistry-10-00201-f001:**
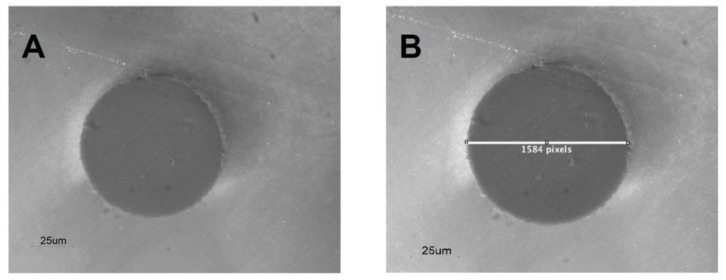
Image of root canal section, which shows the interface between dentin/sealer and sealer/gutta-percha (**A**) and the diameter measurement (**B**).

**Table 1 dentistry-10-00201-t001:** Endodontic sealers and their composition.

Endodontic Sealer	Manufacturer	Composition
AH-Plus^®^	Dentsply Maillefer, Ballaiques, Switzerland	Bisphenol A/F epoxy resin, calcium tungstate, zirconium dioxide, iron oxide pigments dibenzyldiamina, aminoadamantane, silicone oil
GuttaFlow Bioseal^®^	Coltène/Whaledent, GmbH + Co. KG, Germany	Gutta-percha powder, polydimethylsiloxane, platinum catalyst, zirconium dioxide, calcium salicylate, nano-silver particles, coloring, bioactive glass ceramic

**Table 2 dentistry-10-00201-t002:** Infiltration values recorded for each group.

	G1 (AH-Plus^®^, *n* = 15)	G2 (GuttaFlow Bioseal^®^, *n* = 15)	G3 (Negative Control, *n* = 4)	G4 (Positive Control, *n* = 4)
Mean	1.158	0.349	0.331	2.395
Standard deviation	0.768	0.172	0.182	0.234
Minimum value	0.331	0.081	0.167	2.135
Maximum value	3.209	0.718	0.537	2.652

**Table 3 dentistry-10-00201-t003:** Descriptive statistic for both obturation materials (*n* = 72).

Variable	Unit	Amplitude		Median	Mean			SD
		Minimum	Maximum		Value	SEM	95%CI	
G1 (AH-Plus^®^, *n* = 36)								
F	N	0.03	10.05	3.43	3.40	0.40	(2.72 to 4.16)	2.40
Radius	mm	0.026	0.060	0.042	0.042	0.001	(0.039 to 0.044)	0.008
radius	mm	0.023	0.055	0.039	0.039	0.001	(0.036 to 0.042)	0.008
Thickness	mm	0.520	2.300	0.640	0.772	0.064	(0.652 to 0.902)	0.386
Tension	MPa	0.180	93.621	15.181	20.244	3.096	(14.862 to 26.393)	18.579
G2 (GuttaFlow Bioseal^®^, *n* = 36)								
F	N	0.01	9.20	0.83	1.60	0.39	(0.93 to 2.37)	2.33
Radius	mm	0.025	0.062	0.046	0.043	0.002	(0.040 to 0.047)	0.010
radius	mm	0.023	0.057	0.039	0.039	0.002	(0.036 to 0.042)	0.009
Thickness	mm	0.580	1.960	1.560	1.259	0.081	(1.108 to 1.418)	0.484
Tension	MPa	0.024	19.766	2.560	4.880	0.918	(3.235 to 6.648)	5.510

Abbreviations: F (force); SEM (standard error of the mean); 95% CI (95% confidence interval); SD (standard deviation); N (Newton); mm (millimeter); MPa (MegaPascal). Radius: radius of the root canal in the most coronal part; radius: radius of the root canal in the most apical part.

**Table 4 dentistry-10-00201-t004:** Means and standard deviations by group and Mann–Whitney results regarding the effect of the endodontic sealer.

Dependent Variable		X: Independent Variable (Material)		Mann–Whitney			
Y_i_:	Units	G1 (AH-Plus^®^, *n* = 36)	G2 (GuttaFlow Bioseal^®^, *n* = 36)	Z	*p*	Effect Size	
						Cohen’s r	Qualitative
F	N	3.40 ± 2.40	1.60 ± 2.33	3.886	<0.001	0.458	medium
Radius	mm	0.042 ± 0.008	0.043 ± 0.010	0.568	0.570	0.067	small
radius	mm	0.039 ± 0.008	0.039 ± 0.009	0.073	0.942	0.009	small
Thickness	mm	0.772 ± 0.386	1. 259 ± 0.484	4.509	<0.001	0.531	large
Tension	MPa	20.244 ± 18.579	4.880 ± 5.510	5.068	<0.001	0.597	large

Abbreviation: F (force); N (Newton); mm (millimeter); MPa (MegaPascal). Radius: radius of the root canal in the most coronal part; radius: radius of the root canal in the most apical part.

**Table 5 dentistry-10-00201-t005:** Means and standard deviations by group and zone and results of the comparison of the endodontic sealer.

Dependent variable		X: Independent Variable (Material)		Mann–Whitney	
Y_i_:	Units	G1 (AH-Plus^®^, *n* = 36)	G2 (GuttaFlow Bioseal^®^, *n* = 36)	Z	*p*
Cervical zone (*n* = 24)		*n* = 12	*n* = 12		
F	N	2.67 ± 1.80	2.99 ± 3.44	–0.577	0.564
Radius	mm	0.047 ± 0.007	0.048 ± 0.009	–0.058	0.954
radius	mm	0.045 ± 0.007	0.044 ± 0.007	–0.318	0.751
Thickness	mm	0.613 ± 0.040	1.262 ± 0.541	–4.056	<0.001
Tension	MPa	15.008 ± 10.768	7.511 ± 6.712	–2.021	0.043
Medial zone (*n* = 24)		*n* = 12	*n* = 12		
F	N	2.56 ± 1.95	1.08 ± 1.23	–2.108	0.035
Radius	mm	0.043 ± 0.006	0.050 ± 0.006	–2.570	0.010
radius	mm	0.040 ± 0.006	0.043 ± 0.007	–1.530	0.126
Thickness	mm	0.628 ± 0.051	1.492 ± 0.358	–3.999	<0.001
Tension	MPa	16.875 ± 14.104	2.534 ± 2.780	–3.002	0.003
	N				
Apical zone (*n* = 24)		*n* = 12	*n* = 12		
F	N	4.97 ± 2.69	0.73 ± 0.82	–3.754	<0.001
Radius	mm	0.036 ± 0.007	0.032 ± 0.006	–1.502	0.133
radius	mm	0.032 ± 0.006	0.030 ± 0.005	–1.270	0.204
Thickness	mm	1.075 ± 0.564	1.023 ± 0.453	–0.607	0.544
Tension	MPa	28.849 ± 25.697	4.595 ± 5.495	–3.406	0.001

Abbreviation: F (force); N (Newton); mm (millimeter); MPa (MegaPascal). Radius: radius of the root canal in the most coronal part; radius: radius of the root canal in the most apical part.

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
