# Peer review of "Evaluation of the Sealing Ability and Bond Strength of Two Endodontic Root Canal Sealers: An In Vitro Study"

_dentistry, 2022, doi:10.3390/dj10110201_

Round 1

Reviewer 1 Report

I like the article

It is well structured

page 126: indicate the exact mm the section were made

conclusions answer the aims of this work

Reviewer 2 Report

  1. In the abstract, the author described that samples were thirty-eight anterior teen with single root, but not mention maxillary teeth or mandibular teeth. Their canal morphology was different. Are they oval or round cross section? The author did not clearly state the sample collection in the  materials and methods, either.
  2. According to the experiment design, The G1 and G2 groups were filled with single cone technique. However, most of the dentists use AH-plus sealer with  the warm vertical compaction or lateral condensation technique. They use bioceramic sealer with hydraulic condensation. The sealer thickness of both of them are different. Consequently, the conclusion of this study may not be equal to the real condition in clinic.
  3. Page 3 line 104, in the G3 group, the cones were sectioned with a heated  instruments and condensed with hot plugger. It seems that “plugger “ was appropriate. 
  4. The design of the positive control and negative control group were questionable. The G3 should be the positive control and G1, G2 and G3 were covered with varinish, except the apical 3 mm. The negative control design (G4) may fill with proper filling materials in the prepared canal.  AH-plus, GuttaFlow Bioseal or flowable resin is suitable and completely seal the root. Table 2 showed the negative control (G3) was still have leakage. It means the experiment design was not reasonable.
  5. Table 2 was incorrect. The lower part was duplicated the upper part in Italian style.
  6. Please correct all the p value in the same style. For example, table 2, table 3 and 5 were not consistent. 
  7. Figure 1 showed 25 um. Is it scale?
  8. Table 3,4,5 did not describe the first column clearly. There were “two” Radial and radial. What did they mean? The abbreviation should be defined in the legend. For example, “F”  
  9. Page 7 line 235-237, Studies have shown that epoxy resin-based sealers have greater bond strength to the root canal walls than most endodontic sealers [1,15,31,32]. Some of citations did not have such as conclusion. Please check it. 
  10. There are some similar studies have the same conclusion. Please check it and include in the discussion to support your findings.

Reviewer 3 Report

The paper is scientifically sound. However, few necessary changes are required, which are:

1. Please rewrite the conclusion part of the Abstract. The conclusion part should depict the broader picture, instead of just repeating the findings of the results in the conclusion section again.

2. The introduction part needs to be further improved. The authors may use the following relevant papers:

Al-Kheraif AA, Mohamed BA, Sufyan AO, Khan AA, Divakar DD. Photodynamic therapy and other pretreatment methods on epoxy-based glass fiber post on the push-out bond strength to radicular dentin. Photodiagnosis and Photodynamic Therapy. 2021 Dec 1;36:102526.

Al-Kheraif AA, Mohamed BA, Khan AA, Al-Shehri AM. Role of Riboflavin; Curcumin photosensitizers and Ozone when used as canal disinfectant on push-out bond strength of glass fiber post to radicular dentin. Photodiagnosis and Photodynamic Therapy. 2022 Mar 1;37:102592.

3. In section 3.2, the comma has been used instead of the decimal between the values. While in Table 3, the decimal has been used. Please be consistent.

4. The authors have mentioned the null hypothesis in "Introduction". However, their answer to null hypothesis is not stated? Please mention your answer

5. Please suggest some future direction in the discussion part to carry on related work further.

6. The conclusion section just depicts the results findings. Actually, the conclusion section is intended to make the reader understand why your research should matter to them. Please rewrite this section.

Round 2

Reviewer 2 Report

Please recheck line 32 and 374. The space should be corrected. 

Reviewer 3 Report

The authors have addressed all my comments. No more comments